# Modulation of Canine Gut Microbiota by Prebiotic and Probiotic Supplements: A Long-Term In Vitro Study Using a Novel Colonic Fermentation Model

**DOI:** 10.3390/ani14223342

**Published:** 2024-11-20

**Authors:** Alessandro Gramenzi, Luana Clerico, Benedetta Belà, Meri Di Leonardo, Isa Fusaro, Giulia Pignataro

**Affiliations:** 1Department of Veterinary Medicine, University of Teramo, 64100 Teramo, Italy; agramenzi@unite.it (A.G.); platobio31@gmail.com (B.B.); ifusaro@unite.it (I.F.); 2Independent Researcher, 17100 Savona, Italy; luana.clerico@virgilio.it; 3Azienda Sanitaria Locale di Teramo, 64100 Teramo, Italy; meridileonardo@yahoo.it

**Keywords:** canine gut microbiota, prebiotic, probiotic, colonic fermentation model, microbial shifts, microbial diversity

## Abstract

Gut microbiota significantly influences dogs’ well-being, and several researchers have made strides in characterizing canine gut microbiota composition. Recent research has revealed that the canine gut microbiota is highly impacted by the type of diet. Prebiotics (non-digestible oligosaccharides able to promote the growth of beneficial bacteria) and probiotics (live microorganisms that confer a health benefit by enhancing microbial diversity) can modulate the gut microbiota and promote gastrointestinal health. A growing interest has recently been in combining prebiotics and probiotics (synbiotics) to achieve synergistic effects. The present study aimed to investigate the effects of supplementation with a prebiotic (Microbiotal) and a probiotic (*L. reuteri*) using the fecal microbiota of a healthy canine donor. The investigators used a novel in vitro fermentation platform (SCIME™), which closely mimics the canine gastrointestinal tract, allowing long-term experiments. This study demonstrates the promising modulatory effects of prebiotics and probiotics, especially when combined.

## 1. Introduction

The gastrointestinal tract (GIT) of dogs contains a diverse and complex community of microbes—bacteria, archaea, fungi, protozoa, and viruses—that are integral to digestion, immune function, and metabolic processes [1,2,3]. The composition and balance of these microbes, collectively known as the gut microbiota [4], can be influenced by several factors such as diet, environment, and medication [5].

During the last decades, the mammalian intestinal microbiome has received increasing attention for its importance for health. The composition of the intestinal microbial community has been widely associated with health and disease in humans and animals. Maintaining a healthy gut microbiota is essential for overall canine health, as dysbiosis—an imbalance in the microbial community—can lead to conditions such as chronic enteropathy [3,6,7,8,9,10].

Several studies found that the dog’s GIT hosts a complex ecosystem of several hundred different bacterial genera and possibly more than a thousand bacterial phylotypes [11]. It has been estimated that the gut microbiota may include approximately 10 times more microbial cells than the host cell number and that the microbial gene pool may be 100 times greater than that of the host, as demonstrated in various studies in humans, animal models, and, more recently, dogs and cats [4].

The phyla *Firmicutes*, *Bacteroidetes*, *Proteobacteria*, *Fusobacteria*, and *Actinobacteria* constitute more than 99% of all intestinal bacteria in the dog [11].

The *Firmicutes* phylum is highly represented in the intestine, and the most common classes within it are *Clostridia* and *Bacilli*. The *Clostridia* class is very abundant, and the most present families belong to the clusters IV (*Ruminococcaceae*), XI (*Peptostreptococcaceae*), and XIVa (*Lachnospiraceae*). These clusters have been associated with a healthy gastrointestinal tract, including many short-chain fatty acid-producing bacteria, especially butyrate, including *Ruminococcus* spp., *Faecalibacterium* spp., *Dorea* spp., and *Turicibacter* spp. [4,12,13]. The *Bacilli* class is almost exclusively of the *Lactobacillales*, particularly with the genera *Lactobacillus* and *Streptococcus*.

Other quantitatively significantly represented species belong to the Phylum *Bacteroidetes*. The most critical genera included in this phylum are *Bacteroides* and *Prevotella*.

Phyla *Proteobacteria* and *Actinobacteria* are frequently found in the small intestine but, under physiological conditions, are present only in small quantities.

The *Enterobacteriaceae* family, of which *Escherichia coli* is a member, belongs to the phylum *Proteobacteria*. These facultative anaerobic bacteria can exploit the oxygen in the small intestine. Their high presence is linked with pathological states. The phylum *Actinobacteria* includes the *Corynebacteriaceae* and *Coriobacteriaceae* families [12].

While much research has focused on the human gut microbiome, fewer studies have focused on understanding the canine and feline microbiota [14].

Recent research has allowed the characterization of the canine gut microbiome and revealed that the type of diet highly influences it [5]. This has increasingly focused on the use of dietary interventions, particularly prebiotics and probiotics, to modulate the gut microbiota and promote gastrointestinal health [15,16].

Pre- and probiotics are broadly used in human medicine to preserve or restore a healthy condition [17]. However, the employment of these devices is new in veterinary medicine and pet treatment.

Prebiotics, such as non-digestible oligosaccharides (NDOs), are selectively fermented in the gut, promoting the growth of beneficial bacteria, particularly in the proximal colon where saccharolytic fermentation predominates [16,18,19,20,21,22,23,24,25,26].

The use of probiotics is more ancient than that of prebiotics. Probiotics, live microorganisms that confer a health benefit when administered adequately [27], can transiently modify the gut microbiota, enhancing microbial diversity and functional capacity. The use of probiotics in livestock is broadly spread, whereas pet nutrition is still developing. Sauter et al. [28] conducted an ex vivo study in dogs with chronic enteropathies. This research suggested the positive effect of a probiotic multistrain containing three different *Lactobacillus* strains on cytokine expression, mainly through the regulation of T-cells. Rossi et al. [29] found that the administration of probiotics promotes the intensification of T-cell expression. The adjuvant effects of *Enterococcus faecium* have been demonstrated at both intestinal and systemic levels. These effects may be relevant to improving the protective immune response during the weaning period [30].

In recent times, there has been a growing interest in combining prebiotics and probiotics (synbiotics) to achieve synergistic effects [31,32], promoting both microbial growth and metabolic activity [2,15,33,34,35] even if the in vitro assays related to gut health in companion animals were restricted to the use of short-term batch experiments, creating considerable bias as they are often not representative of the in vivo situation.

The present study aimed to demonstrate the benefic effects of a prebiotic (Microbiotal, NBF Lanes, Milan, Italy) [36], a probiotic (*L. reuteri*, NBF Lanes, Milan, Italy) [37,38], and their combination on the gut microbiota of a healthy canine donor. Using the SCIME™ in vitro fermentation platform [39,40,41], which closely mimics canine gastrointestinal conditions, it was possible to explore how these dietary supplements modulate the microbial composition and diversity in both the luminal and mucosal regions of the proximal and distal colon with a long-term experiment. This study provides realistic insights into microbial modulation, underlining the potential of these supplements to enhance gut health in dogs.

## 2. Materials and Methods

### 2.1. Experimental Design

The SCIME™ system, consisting of compartments representing the stomach, small intestine, proximal colon, and distal colon (Figure 1), was employed to simulate the canine GIT.

### 2.2. SCIME^TM^ Technology Platform Adaptation

The SCIME™ model was adapted from the Simulator of the Human Intestinal Microbial Ecosystem (SHIME^®^) [42,43,44] to closely mimic canine gastrointestinal conditions. Specific adaptations included the following:Maintaining the system at a constant body temperature of 39 °C (reflecting canine physiology)Adjusting the pH and retention times in each compartment to match those of the canine gut, with pH levels in the proximal colon set at 5.7–5.9 and in the distal colon at 6.6–6.9Inoculating the system with fresh canine fecal microbiota, which allowed the model to simulate the dynamic changes in microbial composition under controlled conditions.

The system incorporated both luminal and mucosal environments (M-SCIME), which is particularly important as mucosa-associated microbiota play a significant role in barrier function and immune modulation [40,45,46]. Including the mucosa compartment increases the SCIME™’s value and modeling capacity. It allows for evaluating whether a specific treatment can modulate the mucosa-associated microbial community.

To optimally address the research questions for this particular study, the SCIME setup was adapted to a Triple-M-SCIME™ configuration (Figure 2), allowing the comparison of three different conditions in parallel. During this specific project, the properties of three different test ingredients were evaluated using the microbiota of a healthy ± 20 kg canine donor. In practice, each segment of the TripleSCIME^®^ consisted of a succession of three reactors simulating the different parts of the gastrointestinal tract. The first reactor (St + SI) simulated the stomach and the small intestine. The colonic reactor compartments were continuously stirred reactors with constant volume and pH control. Six colon vessels were thus used to assess the effect of three conditions. In practice, each unit consisted of a proximal (pH 5.7–5.9) and distal (pH 6.6–6.9) colon compartment.

The system was inoculated with fresh fecal microbiota obtained from a healthy 20 kg dog donor. This donor was chosen to reflect a typical adult canine, and its microbiota provided a representative model for testing the effects of the dietary supplements. The experimental timeline was divided into three distinct phases: stabilization, control, and treatment periods, each designed to evaluate the microbiota’s baseline behavior and its response to dietary interventions.

**Stabilization period (3 weeks):** This period allowed the microbial community to adapt to the SCIME™ environment, ensuring that the microbial populations stabilized in both the proximal and distal colon compartments before any treatment was introduced.**Control period (2 weeks):** Baseline microbial composition and metabolic activity were measured under normal dietary conditions (without supplements). Samples were collected from both luminal and mucosal environments in the proximal and distal colon to establish a reference point for comparison during the treatment phase.**Treatment period (2 weeks):** Three distinct test products were evaluated:
○**Microbiotal (M)**: A **prebiotic** (one tablet/day containing 865.3 mg of active ingredients such as Oligofructose (FOS) and Inulin as prebiotic fibers; Microencapsulated Tributyrate as the postbiotic; *Lactobacillus reuteri* NBF1 thermally inactivated)○***Lactobacillus reuteri* (P):** A **probiotic** (containing the bacterial strain *Lactobacillus reuteri* DSM 32203) administered at a dose of 2 × 10^10^ CFU/day○**Combined prebiotic and probiotic supplementation (M + P)**: Both products were co-administered to evaluate potential synergistic effects.


Throughout the treatment period, the standard SCIME™ nutrient matrix was supplemented with the respective test products. Samples were collected three times weekly from both the proximal and distal colon.

### 2.3. Microbial Analysis

Microbial composition was analyzed using 16S-targeted Illumina sequencing, a highly sensitive molecular technique that provides detailed insights into the microbial community at multiple taxonomic levels, from phylum to genus. Samples from the luminal and mucosal compartments were collected and subjected to 16S-targeted Illumina sequencing, focusing on two hypervariable regions (V3–V4) of the 16S rDNA [47].

Complementary techniques were employed to analyze microbial shifts [47]:**Quantitative analysis** of microbial populations using flow cytometry was used, which allowed for accurate measurement of total bacterial counts.**Alpha-diversity indices**, including the Chao1, Shannon, and Simpson indices, were used to evaluate microbial richness and evenness within samples.**Beta diversity** was assessed using the discriminant analysis of principal components (DAPC), which provided insight into how microbial communities diverged between control and treatment conditions [48].

### 2.4. Description of Statistics

LEfSe (Linear discriminant analysis effect size) [49] and treeclimbR [50] were used to identify the specific taxa that contributed to significant differences between control and treatment groups. LEfSe enables the identification of bacterial taxa with the most significant effect size (LDA score > 2), while treeclimbR assesses the statistical significance and biological relevance of taxa. These analyses helped to pinpoint the critical microbial groups driving the observed shifts in community composition.

## 3. Results

### 3.1. Alpha and Beta Diversity

#### 3.1.1. Alpha Diversity

Alpha diversity was assessed using several metrics, including observed taxa richness, Chao1, Shannon, and Simpson indices. These metrics were calculated for each treatment in the luminal and mucosal environments of the proximal and distal colon to determine how the different supplements affected microbial richness and evenness (Figure 3, Figure 4, Figure 5 and Figure 6).

**Observed taxa**: After supplementation with each product, an increase in observed taxa was noted. The combination treatment of Microbiotal and *L. reuteri* (M + P) produced the highest increase in observed taxa, particularly in the proximal colon’s luminal environment.**Shannon and Simpson Indices**: Considering species richness and evenness, both indices demonstrated increased microbial diversity in the proximal and distal colon after treatment with all test products. The Shannon index, which places more weight on richness, showed that the combinatory treatment (M + P)t had the most pronounced effect on microbial diversity, particularly in the mucosal environment of the proximal colon. The Simpson index, which emphasizes evenness, also highlighted a more balanced microbial community after the M + P treatment.

##### Beta Diversity

Beta diversity, which reflects differences in microbial community composition between treatments, was evaluated using the discriminant analysis of principal Components (DAPC) to visualize clustering patterns and shifts in microbial populations (Figure 7, Figure 8, Figure 9 and Figure 10).

**Proximal colon**: The DAPC showed distinct clustering for the combination treatment in the luminal environment of the proximal colon, with a significant shift away from the control group. This distinct separation indicates that M + P supplementation notably enhanced the growth of specific beneficial taxa such as *Limosilactobacillus* and *Faecalibacterium*.**Distal colon**: A similar clustering pattern was observed in the distal colon, though the microbial shifts were less pronounced compared to the proximal colon. The most significant change was again seen with the M + P treatment, which induced a notable divergence in both the luminal and mucosal communities from the control group. This highlights the potential for synbiotic supplements to impact microbial composition.

##### Taxonomy Assignment

The luminal and mucosal microbiota in both the proximal and distal colon exhibited significant shifts in response to the different test products. 16S-targeted Illumina sequencing provided insights into changes at various phylogenetic levels, including the phylum (Figure 11, Figure 12, Figure 13 and Figure 14), family (Figure 15, Figure 16, Figure 17 and Figure 18), and genus (Figure 19, Figure 20, Figure 21 and Figure 22), with specific microbial groups enriched or reduced depending on the treatment.

Furthermore, Annex I (provided as Appendix A) gives an overview of community composition for the different test conditions (control and treatments) and for the different taxa (reporting the absolute abundances for the luminal environment and the relative abundance for the mucosal environment).

##### Luminal Microbiota Composition


**Proximal Colon**



**Prebiotic (Microbiotal; M) Treatment**


Treatment with Microbiotal resulted in notable changes in the luminal microbiota composition, particularly in the proximal colon, where saccharolytic fermentation is most active. Significant enrichment of *Bifidobacterium* was observed, consistent with prebiotics’ role in promoting the growth of saccharolytic bacteria capable of fermenting complex carbohydrates into short-chain fatty acids (SCFAs) like acetate and lactate. In addition to *Bifidobacterium*, the genus *Prevotella* was also significantly enriched. This genus is known for deleting plant-derived polysaccharides and producing acetate, which can be cross-fed to other bacterial groups, such as butyrate producers.

2.
**Probiotic (*Lactobacillus reuteri*; P) Treatment**


The administration of *L. reuteri* led to a strong enrichment of the *Limosilactobacillus* genus, particularly OTU 14, which is closely related to *L. reuteri*. This enrichment was especially pronounced in the proximal colon, where *Limosilactobacillus* became one of the dominant taxa. The probiotic treatment also significantly increased the abundance of *Pseudomonas*, *Stenotrophomonas*, and *Faecalibacterium*. The rise in *Faecalibacterium* suggests that the probiotic could enhance butyrate production.

3.
**Combination (Microbiotal + *L. reuteri*; M + P) Treatment**


The combining of Microbiotal and *L. reuteri* resulted in synergistic effects, leading to significant changes in the proximal colon’s luminal microbiota. A marked increase in *Limosilactobacillus* and *Bifidobacterium* was observed, indicating that the prebiotic enhanced the probiotic’s engraftment and promoted the growth of beneficial saccharolytic bacteria. Additionally, the *Lachnospiraceae* family, known for its butyrate-producing capabilities, was significantly enriched in the combination treatment. Interestingly, while the probiotic alone enriched *Faecalibacterium*, the combination treatment further boosted this genus, underscoring the potential for co-supplementation to stimulate butyrate production in the proximal colon. Both *Faecalibacterium* and *Lachnospiraceae*, in higher abundances, highlight the metabolic interactions between different bacterial species, promoting a more functional microbial community.

##### Distal Colon


**Prebiotic (Microbiotal; M) Treatment**


In the distal colon, where fermentation activity is slower, Microbiotal supplementation significantly enriched *Bifidobacterium* and *Prevotella*. The presence of these genera in the distal colon indicates that the prebiotic promoted the degradation of more complex carbohydrates, which persisted into the distal regions of the colon.

2.
**Probiotic (*Lactobacillus reuteri*; P) Treatment**


Similar to the proximal colon, *Limosilactobacillus* was enriched in the distal colon after probiotic treatment. However, the overall effect was less pronounced, as the distal colon generally supports slower microbial growth and fermentation. Still, the probiotic appeared to support the growth of butyrate-producing bacteria, with increases in Alloprevotella and *Peptoclostridium.*

3.
**Combination (Microbiotal + *L. reuteri*; M + P) Treatment**


The combination treatment profoundly affected the distal colon microbiota, significantly enriching *Bifidobacterium* and *Lachnospiraceae*. These changes suggest that M + P supplementation can enhance saccharolytic fermentation even in the distal regions of the colon. Enriching *Lachnospiraceae* in both the proximal and distal colon underscores the potential for synbiotic supplements to enhance butyrate production throughout the gut.

#### 3.1.2. Mucosal Microbiota Composition

##### Proximal Colon


**Prebiotic (Microbiotal; M) Treatment**


Microbiotal supplementation significantly enriched *Bifidobacterium* in the proximal colon mucosa, reflecting the prebiotic’s ability to stimulate beneficial bacteria that adhere to the gut lining. Additionally, the *Actinobacteria* phylum, which includes *Bifidobacterium*, was also enriched in the mucosal environment, further supporting the role of Microbiotal in promoting beneficial microbial populations that play a role in maintaining gut homeostasis.

2.
**Probiotic (*Lactobacillus reuteri*; P) Treatment**


The probiotic treatment significantly enriched *Limosilactobacillus* in the proximal colon mucosa, mirroring the effects observed in the luminal environment. The presence of *Limosilactobacillus* in the mucosal layer is significant, as these bacteria can interact directly with the host’s immune cells.

3.
**Combination (Microbiotal + *L. reuteri*; M + P) Treatment**


The combination of Microbiotal and *L. reuteri* (M + P) produced the most pronounced changes in the proximal colon mucosa, significantly enriching *Faecalibacterium* and *Bifidobacterium*. The presence of *Faecalibacterium* in the mucosal environment suggests that the synbiotic approach may enhance butyrate production.

##### Distal Colon


**Prebiotic (Microbiotal; M) Treatment**


In the distal colon mucosa, Microbiotal supplementation again enriched *Bifidobacterium* and *Actinobacteria*, indicating that the prebiotic promotes beneficial microbial populations throughout the colon, not just in the luminal environment.

2.
**Probiotic (*Lactobacillus reuteri*; P) Treatment**


Following P supplementation, *Limosilactobacillus* was also enriched in the distal colon mucosa, though the effect was less pronounced than in the proximal colon. *Limosilactobacillus* in the distal colon suggests that the probiotic may have long-lasting effects on the mucosal microbiota.

3.
**Combination (Microbiotal + *L. reuteri*; M + P) Treatment**


The M + P treatment significantly enriched *Faecalibacterium* and *Bifidobacterium* in the distal colon mucosa, reflecting the synergistic effects of co-supplementation. The enrichment of butyrate-producing bacteria in the mucosal layer of the distal colon suggests that synbiotic supplementation could have important implications for reducing inflammation and promoting gut health.

## 4. Discussion

Our findings indicate the distinct and complementary beneficial effects of prebiotic (Microbiotal; M) [36] and probiotic (*L. reuteri*; P) [37,38] and their combination (M + P) on the composition of the canine gut microbiota. We used the SCIME™ [39,40,41] model, a novel and sophisticated in vitro fermentation model that closely simulates the canine gastrointestinal tract, to demonstrate how each supplement influenced specific bacterial groups and overall microbial diversity. The uniqueness of this study is also related to the fact that, with this complex platform, we could analyze the microbial shift in both the luminal and mucosal environments of the proximal and distal colon.

In addition, this unique technique can perform dietary treatment experiments for a long time (two weeks). Our findings are similar to those of previous in vitro studies that aimed to improve the health of the canine gastrointestinal tracts. However, all those attempts were fundamentally different by the short-term period of the supplementation treatment [15,33,35].

The prebiotic (Microbiotal) significantly enriched saccharolytic bacteria, particularly *Bifidobacterium* and *Prevotella*, in both the proximal and distal colon [21,23]. The role of these genera in carbohydrate metabolism is well-documented, with *Bifidobacterium* being recognized for its fiber-degrading capabilities and production of short-chain fatty acids (SCFAs), such as acetate and lactate. These metabolites are essential for promoting cross-feeding interactions within the gut microbiota, ultimately driving the production of butyrate, an essential energy source for colonocytes and an anti-inflammatory metabolite [10,51].

The increase in *Prevotella* in the proximal colon further suggests that Microbiotal supplementation enhances the microbial community’s ability to ferment plant-based carbohydrates, generating acetate and other intermediates for cross-feeding bacteria. *Prevotella* species are also associated with the production of succinate, another metabolite that can serve as a substrate for butyrate-producing bacteria. These findings highlight the ability of prebiotics to support a healthy microbial ecosystem by promoting metabolic flexibility and enhancing microbial interactions critical for gut health.

Additionally, the enrichment of *Bifidobacterium* in the distal colon, where fermentation typically slows, indicates that Microbiotal exerts sustained effects on microbial fermentation throughout the entire gut. This effect has important implications for maintaining gut health in the distal colon, a region commonly associated with chronic conditions such as ulcerative colitis. Beneficial saccharolytic bacteria in this region may help reduce inflammation and support the gut barrier.

The probiotic *L. reuteri* demonstrated robust engraftment in both the luminal and mucosal environments of the proximal and distal colon, particularly enriching the *Limosilactobacillus* genus, to which *L. reuteri* belongs. This finding confirms that the probiotic could colonize the gut and integrate into the microbial community. Moreover, the increase in *Limosilactobacillus* is associated with beneficial effects such as lactic acid production and modulation of immune responses. Lactic acid can lower gut pH, creating an unfavorable environment for pathogenic bacteria while promoting the growth of other beneficial microbial groups.

Interestingly, *L. reuteri* supplementation also led to the enrichment of *Pseudomonas*, *Stenotrophomonas*, and *Faecalibacterium*, which indicates that the probiotic stimulates interactions with other bacterial groups, potentially supporting cross-feeding dynamics. The increase in *Faecalibacterium*, a well-known butyrate producer, is particularly significant, as butyrate plays a critical role in maintaining gut barrier function, reducing inflammation, and supporting immune health [51].

The presence of these beneficial bacterial groups underscores the potential of probiotics to modulate microbial communities beyond their direct engraftment, promoting a healthier and more resilient gut ecosystem [14,27].

One of the most critical findings from this study is the synergistic effect observed when Microbiotal and *L. reuteri* were co-administered. The combination treatment consistently outperformed the individual supplements in terms of microbial diversity, SCFA production potential, and beneficial bacterial abundance. This benefit was particularly evident in both the proximal and distal colon, where *Bifidobacterium*, *Limosilactobacillus*, and butyrate-producing genera such as *Faecalibacterium* and *Lachnospiraceae* were significantly enriched. The combination treatment enhanced the fermentation of carbohydrates and reduced the abundance of potentially harmful taxa, further promoting a healthier gut environment.

The combination of prebiotic and probiotic supplements enhanced the fermentation of carbohydrates, as evidenced by the increased abundance of saccharolytic bacteria and butyrate producers. Butyrate is a critical metabolite in gut health, serving as a primary energy source for colonic epithelial cells, exerting anti-inflammatory effects, and modulating immune responses [3,10]. Enriching both *Faecalibacterium* and *Lachnospiraceae*, the synbiotic approach likely promoted higher butyrate levels, which could have important therapeutic implications for managing inflammatory conditions such as IBD.

Furthermore, the combination treatment reduced the abundance of *Megamonas*, a genus often associated with dysbiosis in several gastrointestinal disorders. This advantage suggests that synbiotic intervention may help restore microbial balance and reduce the prevalence of potentially harmful taxa, further promoting a healthier gut environment [26].

The results of this study provide strong evidence that both prebiotics and probiotics, alone and in combination, have the potential to modulate the canine gut microbiota beneficially. Enriching beneficial bacteria, including *Bifidobacterium*, *Limosilactobacillus*, and *Faecalibacterium*, suggest that these supplements are essential in maintaining gut health and preventing or managing dysbiosis-related conditions. These findings pave the way for the development of new dietary interventions that leverage the synergistic effects of prebiotics and probiotics to promote a diverse, balanced, and functionally robust microbial community in dogs.

Prebiotics’ ability to enhance saccharolytic fermentation and support butyrate production throughout the colon is particularly noteworthy. This ability could help prevent or mitigate inflammation in the distal colon, a region often linked with chronic gastrointestinal diseases. By promoting the growth of beneficial bacterial populations and enhancing the production of health-promoting metabolites such as butyrate, prebiotics like Microbiotal may serve as valuable dietary interventions for dogs with gastrointestinal issues or those at risk of developing such conditions.

The probiotic *L. reuteri*, through its direct colonization and promotion of cross-feeding interactions, also holds promise for modulating the canine gut microbiota. The significant increase in *Faecalibacterium* following probiotic treatment highlights its potential to support butyrate production and reduce inflammation, which could have important implications for managing conditions like IBD or chronic diarrhea in dogs.

The main implication of this study indicates to the colleagues that, during their daily practice, a synbiotic (prebiotic + probiotic) intervention may offer a superior approach to modulating the canine gut microbiota compared to prebiotics or probiotics alone. The synergistic effects observed with the combination of Microbiotal and *L. reuteri* point to the potential for these supplements to work together in promoting a diverse, balanced, and functionally robust microbial community. This optimistic outlook could be particularly beneficial for dogs suffering from chronic gastrointestinal disorders or those recovering from antibiotic treatment, which often disrupts the gut microbiota [14,31].

However, while this study provides valuable insights using an in vitro model, future in vivo studies should be considered to confirm these findings and assess the long-term effects of prebiotic and probiotic supplementation in live animals. Studies focusing on the impact of these supplements on gastrointestinal health markers, immune responses, and clinical outcomes in dogs with dysbiosis or GI diseases will be critical for translating these findings into clinical practice.

Additionally, the potential for these supplements to prevent gastrointestinal disorders in healthy dogs by promoting resilient and balanced gut microbiota should also be explored. Longitudinal studies investigating the prophylactic use of synbiotics could help elucidate their role in maintaining gut health over time, particularly in dogs exposed to stressors such as dietary changes, environmental shifts, or medication use.

## 5. Conclusions

This study demonstrates the modulatory effects of prebiotics (Microbiotal) and probiotics (*L. reuteri*) and their combination on the canine gut microbiota using an in vitro model that closely simulates the gastrointestinal environment. The combination of prebiotic and probiotic supplements produced the most pronounced shifts in microbial composition, promoting beneficial bacterial populations involved in saccharolytic fermentation and butyrate production. These results suggest that combined prebiotic and probiotic supplementation may significantly benefit canine gut health by enhancing microbial diversity, offering hope for improved treatments in the future. Further in vitro studies are warranted to explore the metabolic activity of the gut microbiota using the same combination of supplements and the same novel platform.

## Figures and Tables

**Figure 1 animals-14-03342-f001:**
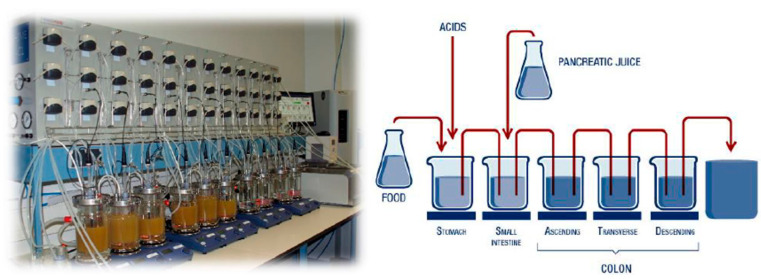
Standard setup of the Simulator of the Canine Intestinal Microbial Ecosystem (SCIME™), consisting of four sequential reactors, simulating the different canine gastrointestinal tract regions.

**Figure 2 animals-14-03342-f002:**
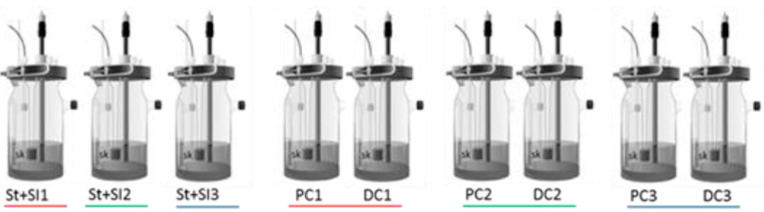
Modified version of the SCIME into a Triple-M-SCIME used for the current study. St + SI: vessel serving as stomach and small intestine, PC: proximal colon, and DC: distal colon.

**Figure 3 animals-14-03342-f003:**
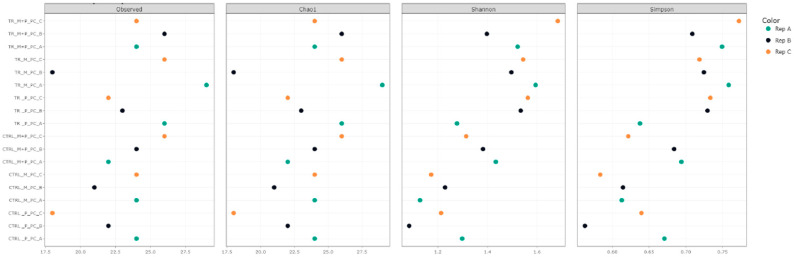
Effect of treatment with the different test products (Microbiotal, M; probiotic, P; and their combination, M + P) on alpha diversity as calculated using four different measures (observed (count of unique taxa in each sample), Chao1, Shannon, and Simpson) in the luminal proximal colon (PC) at the end of the control (CTRL) and treatment (TR) period. Three samples (A, B, C) were collected during each period, represented by different colors.

**Figure 4 animals-14-03342-f004:**
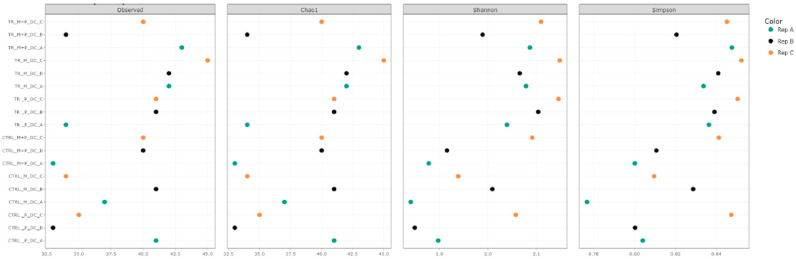
Effect of treatment with the different test products (Microbiotal, M; probiotic, P; and their combination, M + P) on alpha diversity as calculated using four different measures (observed (count of unique taxa in each sample), Chao1, Shannon, and Simpson) in the luminal distal colon (DC) at the end of the control (CTRL) and treatment (TR) period. Three samples (A, B, C) were collected during each period, represented by different colors.

**Figure 5 animals-14-03342-f005:**
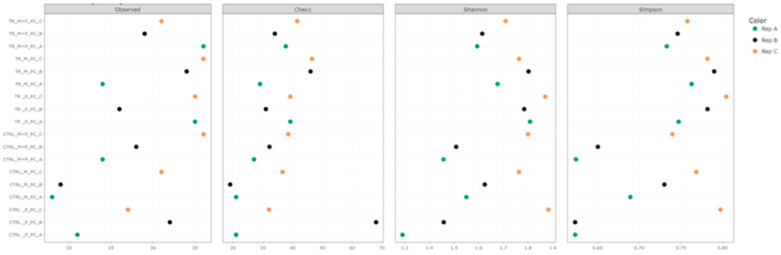
Effect of treatment with the different test products (Microbiotal, M; probiotic, P; and their combination, M + P) on alpha diversity as calculated using four different measures (observed (count of unique taxa in each sample), Chao1, Shannon, and Simpson) in the mucosal proximal colon (PC) at the end of the control (CTRL) and treatment (TR) period. Three samples (A, B, C) were collected during each period, represented by different colors.

**Figure 6 animals-14-03342-f006:**
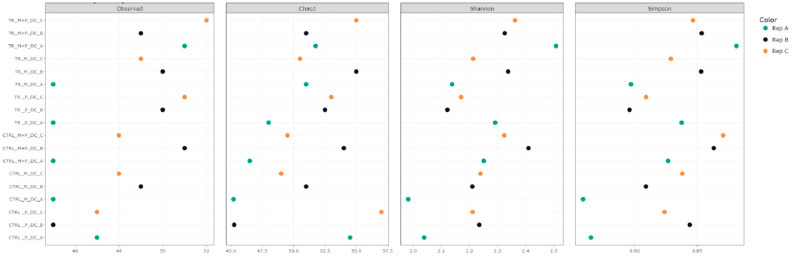
Effect of treatment with the different test products (Microbiotal, M; probiotic, P; and their combination, M + P) on alpha diversity as calculated using four different measures (observed (count of unique taxa in each sample), Chao1, Shannon, and Simpson) in the mucosal distal colon (DC) at the end of the control (CTRL) and treatment (TR) period. Three samples (A, B, C) were collected during each period, represented by different colors.

**Figure 7 animals-14-03342-f007:**
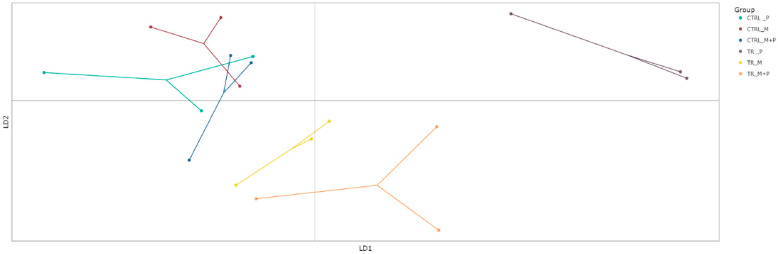
Discriminant analysis of principal components (DAPC) to show differences in community composition (beta diversity) in the luminal proximal colon (PC) at the end of the control (CTRL) and treatment (TR) period following treatment with the different test products (Microbiotal, M; probiotic, P; and their combination, M + P). Each color represents one of six categories (groups), i.e., CTRL_M (n = 3), TR_M (n = 3), CTRL_P (n = 3), TR_P (n = 3), CTRL_M + P (n = 3), and TR_M + P (n = 3). Each dot represents one sample.

**Figure 8 animals-14-03342-f008:**
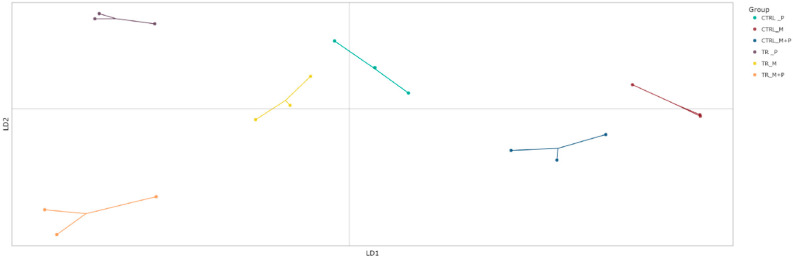
Discriminant analysis of principal components (DAPC) to show differences in community composition (beta diversity) in the luminal distal colon (DC) at the end of the control (CTRL) and treatment (TR) period following treatment with the different test products (Microbiotal, M; probiotic, P; and their combination, M + P). Each color represents one of six categories (groups), i.e., CTRL_M (n = 3), TR_M (n = 3), CTRL_P (n = 3), TR_P (n = 3), CTRL_M + P (n = 3), and TR_M + P (n = 3). Each dot represents one sample.

**Figure 9 animals-14-03342-f009:**
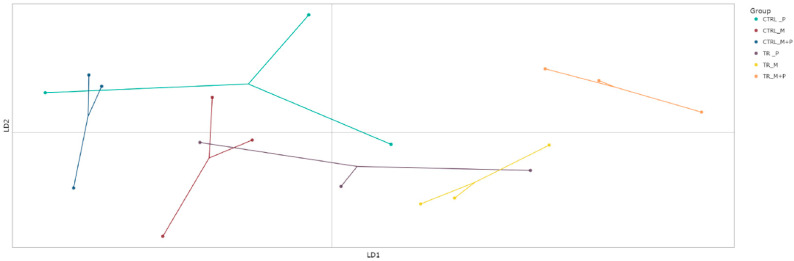
Discriminant analysis of principal components (DAPC) to show differences in community composition (beta diversity) in the mucosal proximal colon (PC) at the end of the control (CTRL) and treatment (TR) period following treatment with the different test products (Microbiotal, M; probiotic, P; and their combination, M + P). Each color represents one of six categories (groups), i.e., CTRL_M (n = 3), TR_M (n = 3), CTRL_P (n = 3), TR_P (n = 3), CTRL_M + P (n = 3), and TR_M + P (n = 3). Each dot represents one sample.

**Figure 10 animals-14-03342-f010:**
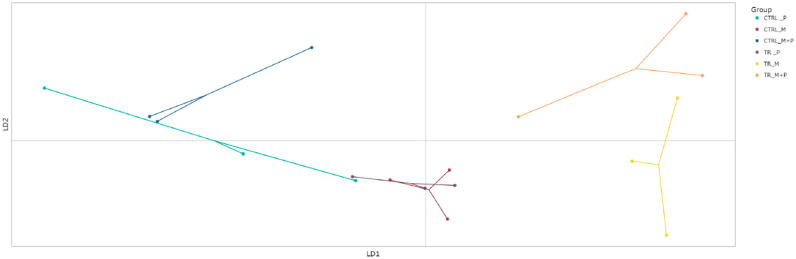
Discriminant analysis of principal components (DAPC) to show differences in community composition (beta diversity) in the mucosal distal colon (DC) at the end of the control (CTRL) and treatment (TR) period following treatment with the different test products (Microbiotal, M; probiotic, P; and their combination, M + P). Each color represents one of six categories (groups), i.e., CTRL_M (n = 3), TR_M (n = 3), CTRL_P (n = 3), TR_P (n = 3), CTRL_M + P (n = 3), and TR_M + P (n = 3). Each dot represents one sample.

**Figure 11 animals-14-03342-f011:**
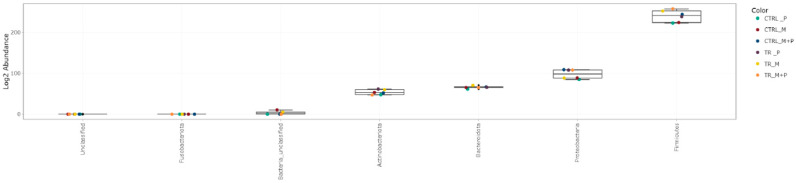
Jitter plots showing abundances of different phyla in the luminal proximal colon (PC) following treatment with the different test products (Microbiotal, M; probiotic, P; and their combination, M + P) at the end of the control (CTRL) and treatment (TR) period based on absolute levels. Each color represents one of six categories (groups), i.e., CTRL_M (n = 3), TR_M (n = 3), CTRL_P (n = 3), TR_P (n = 3), CTRL_M + P (n = 3), and TR_M + P (n = 3).

**Figure 12 animals-14-03342-f012:**
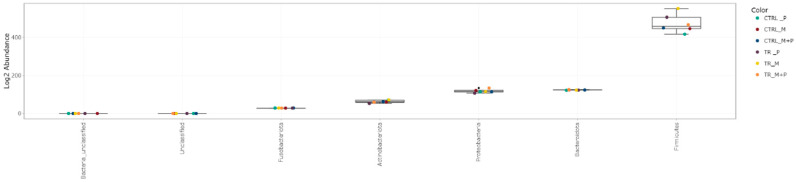
Jitter plots showing abundances of different phyla in the luminal distal colon (DC) following treatment with the different test products (Microbiotal, M; probiotic, P; and their combination, M + P) at the end of the control (CTRL) and treatment (TR) period based on absolute levels. Each color represents one of six categories (groups), i.e., CTRL_M (n = 3), TR_M (n = 3), CTRL_P (n = 3), TR_P (n = 3), CTRL_M + P (n = 3), and TR_M + P (n = 3).

**Figure 13 animals-14-03342-f013:**
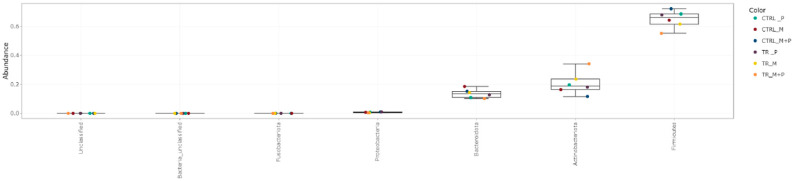
Jitter plots showing abundances of different phyla in the mucosal proximal colon (PC) following treatment with the different test products (Microbiotal, M; probiotic, P; and their combination, M + P) at the end of the control (CTRL) and treatment (TR) period based on relative abundances. Each color represents one of six categories (groups), i.e., CTRL_M (n = 3), TR_M (n = 3), CTRL_P (n = 3), TR_P (n = 3), CTRL_M + P (n = 3), and TR_M + P (n = 3).

**Figure 14 animals-14-03342-f014:**
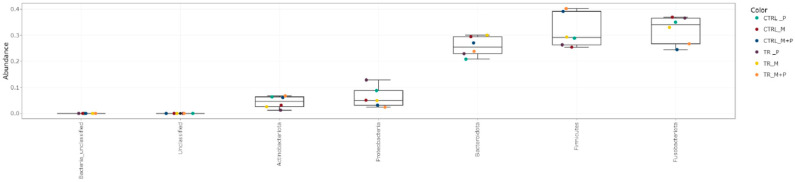
Jitter plots showing abundances of different phyla in the mucosal distal colon (DC) following treatment with the different test products (Microbiotal, M; probiotic, P; and their combination, M + P) at the end of the control (CTRL) and treatment (TR) period based on relative abundances. Each color represents one of six categories (groups), i.e., CTRL_M (n = 3), TR_M (n = 3), CTRL_P (n = 3), TR_P (n = 3), CTRL_M + P (n = 3), and TR_M + P (n = 3).

**Figure 15 animals-14-03342-f015:**
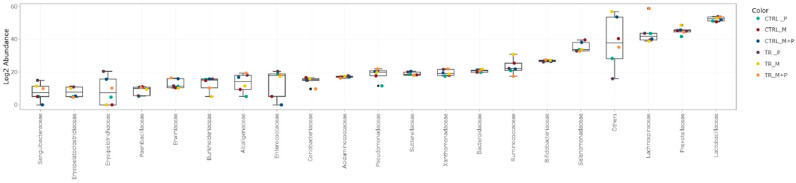
Jitter plots showing abundances of the 20 most abundant families in the luminal proximal colon (PC) following treatment with the different test products (Microbiotal, M; probiotic, P; and their combination, M + P) at the end of the control (CTRL) and treatment (TR) period based on absolute levels. Each color represents one of six categories (groups), i.e., CTRL_M (n = 3), TR_M (n = 3), CTRL_P (n = 3), TR_P (n = 3), CTRL_M + P (n = 3), and TR_M + P (n = 3).

**Figure 16 animals-14-03342-f016:**
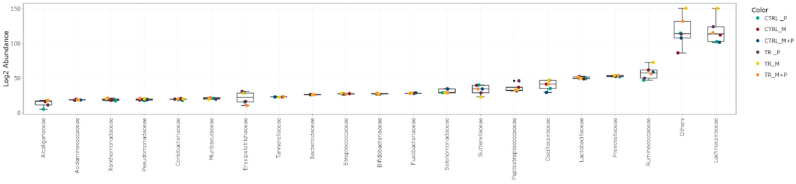
Jitter plots showing abundances of the 20 most abundant families in the luminal distal colon (DC) following treatment with the different test products (Microbiotal, M; probiotic, P; and their combination, M + P) at the end of the control (CTRL) and treatment (TR) period based on absolute levels. Each color represents one of six categories (groups), i.e., CTRL_M (n = 3), TR_M (n = 3), CTRL_P (n = 3), TR_P (n = 3), CTRL_M + P (n = 3), and TR_M + P (n = 3).

**Figure 17 animals-14-03342-f017:**
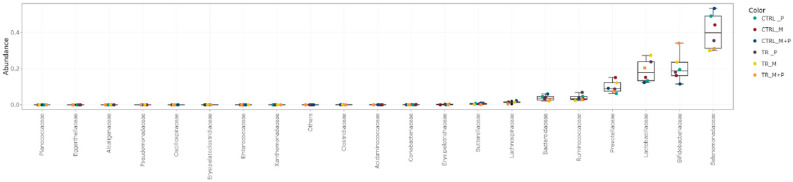
Jitter plots showing abundances of the 20 most abundant families in the mucosal proximal colon (PC) following treatment with the different test products (Microbiotal, M; probiotic, P; and their combination, M + P) at the end of the control (CTRL) and treatment (TR) period based on relative abundances. Each color represents one of six categories (groups), i.e., CTRL_M (n = 3), TR_M (n = 3), CTRL_P (n = 3), TR_P (n = 3), CTRL_M + P (n = 3), and TR_M + P (n = 3).

**Figure 18 animals-14-03342-f018:**
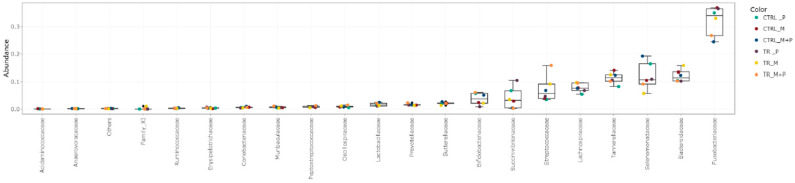
Jitter plots showing abundances of the 20 most abundant families in the mucosal distal colon (DC) following treatment with the different test products (Microbiotal, M; probiotic, P; and their combination, M + P) at the end of the control (CTRL) and treatment (TR) period based on relative abundances. Each color represents one of six categories (groups), i.e., CTRL_M (n = 3), TR_M (n = 3), CTRL_P (n = 3), TR_P (n = 3), CTRL_M + P (n = 3), and TR_M + P (n = 3).

**Figure 19 animals-14-03342-f019:**
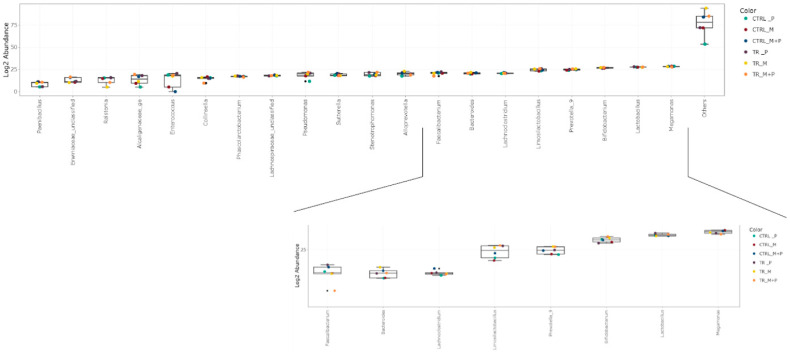
Jitter plots showing abundances of the 20 most abundant genera in the luminal proximal colon (PC) following treatment with the different test products (Microbiotal, M; probiotic, P; and their combination, M + P) at the end of the control (CTRL) and treatment (TR) period based on absolute levels. Each color represents one of six categories (groups), i.e., CTRL_M (n = 3), TR_M (n = 3), CTRL_P (n = 3), TR_P (n = 3), CTRL_M + P (n = 3), and TR_M + P (n = 3).

**Figure 20 animals-14-03342-f020:**
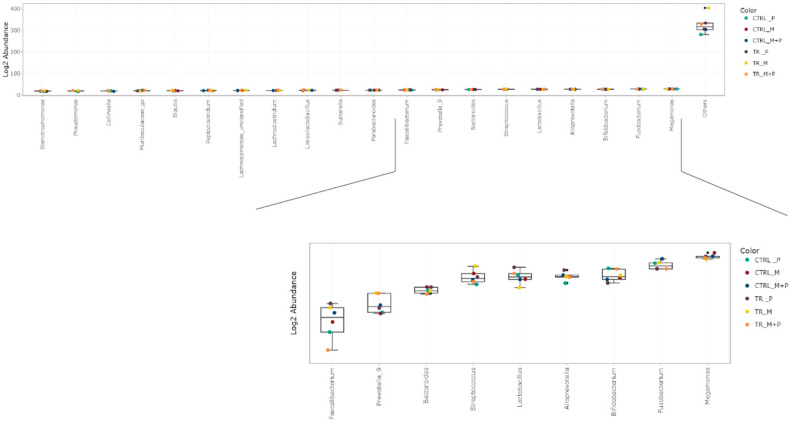
Jitter plots showing abundances of the 20 most abundant genera in the luminal distal colon (DC) following treatment with the different test products (Microbiotal, M; probiotic, P; and their combination, M + P) at the end of the control (CTRL) and treatment (TR) period based on absolute levels. Each color represents one of six categories (groups), i.e., CTRL_M (n = 3), TR_M (n = 3), CTRL_P (n = 3), TR_P (n = 3), CTRL_M + P (n = 3), and TR_M + P (n = 3).

**Figure 21 animals-14-03342-f021:**
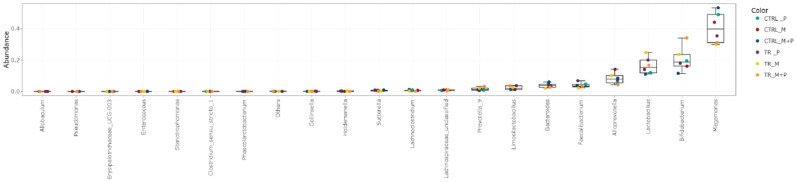
Jitter plots showing abundances of the 20 most abundant genera in the mucosal proximal colon (PC) following treatment with the different test products (Microbiotal, M; probiotic, P; and their combination, M + P) at the end of the control (CTRL) and treatment (TR) period based on relative abundances. Each color represents one of six categories (groups), i.e., CTRL_M (n = 3), TR_M (n = 3), CTRL_P (n = 3), TR_P (n = 3), CTRL_M + P (n = 3), and TR_M + P (n = 3).

**Figure 22 animals-14-03342-f022:**
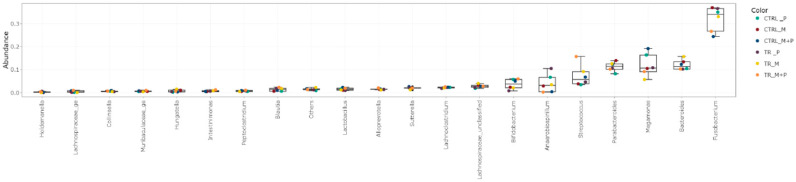
Jitter plots showing abundances of the 20 most abundant genera in the mucosal distal colon (DC) following treatment with the different test products (Microbiotal, M; probiotic, P; and their combination, M + P) at the end of the control (CTRL) and treatment (TR) period based on relative abundances. Each color represents one of six categories (groups), i.e., CTRL_M (n = 3), TR_M (n = 3), CTRL_P (n = 3), TR_P (n = 3), CTRL_M + P (n = 3), and TR_M + P (n = 3).

## Data Availability

The data presented in this study are available upon request from the corresponding authors. The data are not publicly available due to privacy protection.

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
