# Peer review of "Modulation of Canine Gut Microbiota by Prebiotic and Probiotic Supplements: A Long-Term In Vitro Study Using a Novel Colonic Fermentation Model"

_animals, 2024, doi:10.3390/ani14223342_

Round 1

Reviewer 1 Report

Comments and Suggestions for Authors

Overall, the manuscript entitled " Modulation of Canine Gut Microbiota by Prebiotic and Probiotic Supplements: A Long-Term In Vitro Study Using a Novel Colonic Fermentation Model"addresses an interesting and timely topic. Here are my comments and suggestions:

Title, Abstract, and Keywords:

The title relates well with the described work.

The abstract is well organized and elucidative, starting with a justification for the study followed by the brief explanation of the methodology followed in the research. Then shows relevant results obtained and a conclusion remark. Add aim or hipotesis of this study.

The keywords are fine.

The Introduction chapter

The introduction presents some state of the art on a number of topics which are essential to the work that was carried out, and it serves as a justification for its development.

At the end of this chapter there is no clear hypothesis and/or research objective. 

The materials and methods chapter

This part is clear 

The Results chapter

The presentation of results is clear in general.

The discussion chapter

The authors should provide a more detailed discussion of the implications of their results, considering both the strengths and limitations of their study in relation to previous studies.

The conclusions

The authors should revise the conclusions to provide a clear, concise summary of the contribution of the study to the field. Need to be rewritten. An idea may be to synthetize in 3-5 bullet the key results of the study, evidences and recommendation. This improvement will increase clearness and readability. Add a practical implications statement.

Author Response

Dear Reviewer,

Thank you for your comments. We emphasized the strengths, the limitations , and compared with the few previous studies. 

Title, Abstract, and Keywords:

The title relates well with the described work.

The abstract is well organized and elucidative, starting with a justification for the study followed by the brief explanation of the methodology followed in the research. Then shows relevant results obtained and a conclusion remark. Add aim or hipotesis of this study. 

Added

The keywords are fine.

The Introduction chapter

The introduction presents some state of the art on a number of topics which are essential to the work that was carried out, and it serves as a justification for its development.

At the end of this chapter there is no clear hypothesis and/or research objective.

Done by insisting on the research objective. I reinforced the hypothesis.

The materials and methods chapter

This part is clear

The Results chapter

The presentation of results is clear in general.

The discussion chapter

The authors should provide a more detailed discussion of the implications of their results, considering both the strengths and limitations of their study in relation to previous studies.

We emphasized the strengths, the limitations were already there, and compared with the few previous studies (already in the introduction).

The conclusions

The authors should revise the conclusions to provide a clear, concise summary of the contribution of the study to the field. Need to be rewritten. An idea may be to synthetize in 3-5 bullet the key results of the study, evidences and recommendation. This improvement will increase clearness and readability. Add a practical implications statement.

We have summarized the results concisely and clearly, added a few considerations

Reviewer 2 Report

Comments and Suggestions for Authors

Modulation of Canine Gut Microbiota by Prebiotic and Probi-2 otic Supplements: A Long-Term In Vitro Study Using a Novel 3 Colonic Fermentation Model

The authors have conducted interesting and novel research. Some ways to improve the manuscript are listed below:

Line 11 -13 Rephrase due to poor English language

Lines 51-52 instead IBD and chronic diarrhea use the term chronic enteropathy

Also, it has been found that obese dogs differ in their microbiome from lean ones (Handl et al 2013), but microbiome alterations do not lead to obesity (it is more likely that it is the other way around). If a study clearly conducted that research – please provide a citation.

The introduction of the paper can be shorter – text in lines 59-81 can be abbreviated.

Line 95 – employment – replace with use

Line 98 – not a true claim

Line 100 – cocktail – multistrain

Line 101-103 – it seems that 2 sentences are combined by mistake. Reference 29 has no puppies. - Rossi, G.; Pengo, G.; Caldin, M.; Palumbo Piccionello, A.; Steiner, J.; Chen, N.; Jergens, A.; Suchodolski, J. Comparison of micro-668 biological histological, and immunomodulatory parameters in response to treatment with either combination therapy with 669 prednisone and metronidazole or probiotic VSL#3 strains in dogs with idiopathic inflammatory bowel disease. PLoS One. 2014, 670 9, e94699.

Materials and methods: please state clearly how you have divided treatments and how you have named them, how you will sample them in clear and concise manner - Effect of treatment with the different test products (Microbiotal, M; probiotic, P; and their combination, M+P)

Results

Please omit all the discussion from the results. That will lead to the needed shortening of the results section in the manuscript as it is unnecessarily long. Keep it short and concise.

The text on the pictures is too small – please reconsider the different ways of showcasing the graphs to enable better visualization of results.

In the discussion, authors should include more recent studies from 2023 and 2024 (not just the following study

Belà, B.; Crisi, P.; Pignataro, G.; Fusaro, I.; Gramenzi, A. Effects of Nutraceutical Treatment on the Intestinal Microbiota of Sled 686 Dogs. Animals open acc. 2024, 14.

Comments on the Quality of English Language

The English could be improved to more clearly express the research.

Author Response

Dear Reviewer,

Thank you for your comments.

Line 11 -13 Rephrase due to poor English language 

Done

Lines 51-52 instead IBD and chronic diarrhea use the term chronic enteropathy 

Done

Also, it has been found that obese dogs differ in their microbiome from lean ones (Handl et al 2013), but microbiome alterations do not lead to obesity (it is more likely that it is the other way around). If a study clearly conducted that research – please provide a citation. 

Done

The introduction of the paper can be shorter – text in lines 59-81 can be abbreviated. We reduced it

Line 95 – employment – replace with use 

Done

Line 98 – not a true claim  

Done

Line 100 – cocktail – multistrain   

Done

Line 101-103 – it seems that 2 sentences are combined by mistake. Reference 29 has no puppies. - Rossi, G.; Pengo, G.; Caldin, M.; Palumbo Piccionello, A.; Steiner, J.; Chen, N.; Jergens, A.; Suchodolski, J. Comparison of micro-668 biological histological, and immunomodulatory parameters in response to treatment with either combination therapy with 669 prednisone and metronidazole or probiotic VSL#3 strains in dogs with idiopathic inflammatory bowel disease. PLoS One. 2014, 670 9, e94699. 

We have removed the part about puppies

Materials and methods: please state clearly how you have divided treatments and how you have named them, how you will sample them in clear and concise manner - Effect of treatment with the different test products (Microbiotal, M; probiotic, P; and their combination, M+P)
Results  

Done

Please omit all the discussion from the results. That will lead to the needed shortening of the results section in the manuscript as it is unnecessarily long. Keep it short and concise. 

We reduced a lot the results

The text on the pictures is too small – please reconsider the different ways of showcasing the graphs to enable better visualization of results  

We sent sharper image to the editor

In the discussion, authors should include more recent studies from 2023 and 2024 (not just the following study   
Belà, B.; Crisi, P.; Pignataro, G.; Fusaro, I.; Gramenzi, A. Effects of Nutraceutical Treatment on the Intestinal Microbiota of Sled 686 Dogs. Animals open acc. 2024, 14.  

We found no other recent studies on the activity of L. reuteri on the dog

We resolved the points indicated in your requests of elucidation and improved English. Thank you

Reviewer 3 Report

Comments and Suggestions for Authors

A manuscript (animals-3287654) entitled “Modulation of Canine Gut Microbiota by Prebiotic and Probiotic Supplements: A Long-Term In Vitro Study Using a Novel Colonic Fermentation Model” by Alessandro Gramenzi investigates the effects of pre- and probiotic supplements on the intestinal microbiota of a healthy donor dog, using an artificial fermentation platform (in vitro), SCIMETM. The items described in the manuscript meet the scope of the journal Animals, in “Animal diseases and public health”. Fifty-two references were used to construct the article, many of which are from ten years ago, but the vast majority were published in important scientific journals, making the consulted database very strong. The methodology described was meticulously detailed, including very enlightening images. Methodological details were provided, allowing true experimental reproduction by other researchers. The results are presented in a very well explained and detailed manner, with supporting images. In line with this, there are additions to the results that explain the importance of such data. The discussion is clear and objective, providing more detailed explanations of the results, discussing them alongside other studies. The conclusion addresses the topic and highlights the importance of the study.

Author Response

Dear Rewier,

Thank you for your comments and appreciation.